# Burnout among healthcare providers: Its prevalence and association with anxiety and depression during the COVID-19 pandemic in Macao, China

Yu Zheng[1☯], Pou Kuan Tang[1☯], Guohua Lin[1], Jiayu Liu[1], Hao Hu[1,2], Anise Man Sze Wu[3], Carolina Oi Lam Ung[1,2]*

1 State Key Laboratory of Quality Research in Chinese Medicine, Institute of Chinese Medical Science, University of Macau, Macao, China, 2 Faculty of Health Sciences, Department of Public Health and Medicinal Administration, University of Macau, Macao, China, 3 Faculty of Social Sciences, University of Macau, Macao, China

☯ These authors contributed equally to this work.
* carolinaung@um.edu.mo

**Data Availability Statement:** The survey data is available on request provided that approval for sharing the anonymised data is granted by the

## Abstract

### Introduction

Burnout in healthcare providers (HPs) might lead to negative consequences at personal, patient-care and healthcare system levels especially during the COVID-19 pandemic. This study aimed to investigate the prevalence of burnout and the contributing variables, and to explore how, from health workforce management perspective, HPs' experiences related to carrying out COVID-19 duties would be associated with their burnout.

### Methods

A cross-sectional, open online survey, informed by physical and psychological attributes reportedly related to burnout, the Copenhagen Burnout Inventory (CBI) and the Hospital Anxiety and Depression Scale (HADS), was completed by HPs in Macau, China during October and December 2021. Factors associated with burnout were analysed using multiple logistic regressions.

### Results

Among the 498 valid responses, the participants included doctors (37.5%), nurses (27.1%), medical laboratory technologist (11.4%) and pharmacy professionals (10.8%), with the majority being female (66.1%), aged between 25-44years (66.0%), and participated in the COVID-19 duties (82.9%). High levels of burnout (personal (60.4%), work-related (50.6%) and client-related (31.5%)), anxiety (60.6%), and depression (63.4%) were identified. Anxiety and depression remained significantly and positively associated with all types of burnout after controlling for the strong effects of demographic and work factors (e.g. working in the public sector or hospital, or having COVID-19 duties). HPs participated in COVID-19 duties were more vulnerable to burnout than their counterparts and were mostly dissatisfied with

Panel on Research Ethics of the Research Committee, University of Macau. To request such access, please email rskto.ethics@um.edu.mo.

**Funding:** The research was funded by the University of Macau (SRG2021-00007-ICMS). The funders had no role in study design, data collection and analysis, decision to publish, or preparation of the manuscript.

**Competing interests:** This study was conducted in the absence of any commercial or financial relationships that could be construed as a potential conflict of interest.

the accessibility of psychological support at workplace (62.6%), workforce distribution for COVID-19 duties (50.0%), ability to rest and recover (46.2%), and remuneration (44.7%), all of which were associated with the occurrence of burnout.

## Conclusions

Personal, professional and health management factors were found attributable to the burnout experienced by HPs during the COVID-19 pandemic, requiring actions from individual and organizational level. Longitudinal studies are needed to monitor the trend of burnout and to inform effective strategies of this occupational phenomenon.

## Introduction

Three years into the pandemic, COVID-19 continues to evolve, presenting ongoing challenges to the public health response and overwhelming the health systems. Healthcare providers (HPs) are the essential component of any health systems bearing the responsibilities of responding to and controlling the pandemic in the forefront. Working under immense stress, HPs were often required to multi-task and promptly acquire new skillsets to execute various public health measures [1]. Compounded by high workload, demanding working environment and schedule, and infection susceptibility at work over time, HPs can easily become vulnerable to various mental health impact [2–4].

Developed in response to chronic stressors at work, burnout in HPs has been widely recognized as an important occupational hazard during this pandemic [5]. According to the World Health Organization (WHO), burnout was emphasized as one of the key parameters of HPs' health under the impact of COVID-19 [6]. Burnout is defined as "*a syndrome conceptualized as resulting from chronic workplace stress that has not been successfully managed*" [7]. Affected HPs may experience irritability, mood swings, insomnia, symptoms of emotional and physical fatigue, resulting in profound physical and mental adverse health outcomes [8]. A lack of motivation and low morale [9, 10] and an impact on HPs' ability to respond to changes in the clinical setting had been reported about burnout [8, 11]. HPs' compromised professional performance may in turn result in poor or even unsafe outcomes for the patients and imposing immense additional costs to the health system [12, 13].

Although burnout had been an issue for HPs even before the pandemic and studies have shown that viral outbreaks including COVID-19 have exacerbated this further [14–18], the risk and protective factors of burnout are not fully understood. In addition, recent data has drawn further attention to the links between burnout and other mental illnesses such as anxiety and depression [19]. Anxiety and depression at the early phase of the COVID-19 pandemic were common among HPs [20, 21], and suggested to be closely related to burnout [22–27]. Nevertheless, how burnout is associated with anxiety and depression remains unclear [28, 29]. This presents great challenges to addressing the mental health issues associated with the pandemic for HPs. In fact, adequate support HPs' mental health being with respect to the local context has been recommended as a key to maintaining and strengthening of public health capacities [30–33].

Macao, one of the most densely populated places in the world (a population of about 680,000 over an area of 32.9 km$^2$) and a famous tourist destination (nearly 40 million visitors in 2019), has allocated much of the healthcare resources to maintain strict prevention and control measures since the first case of COVID-19 infection reported 22 January 2020 [34, 35]. Frontline HPs, while maintaining essential healthcare services, had to adjust to work towards

the requirements of the evolving public health measures, including but not limited to stringent health checks at points of entry, risk communication, population-wide face mask distribution, operating COVID-19 infectious disease ward and quarantine facilities, surveillance, epidemiological investigation, contact tracing, therapeutic logistics, polymerase chain reaction (PCR) tests, and the COVID-19 vaccination programme. In May 2020, a medical team was even dispatched to support Algeria's fight against COVID-19. Despite the immense stress imposed on the HPs, little is known about the burnout phenomenon among them.

In light of the above, this study aimed to investigate the prevalence of burnout after two years of the COVID-19 outbreak, to investigate the association of burnout with anxiety, depression and other contributing variables, and to explore how, from health workforce management perspective, HPs' experiences related to carrying out COVID-19 duties would be associated with their burnout. It is expected that the study findings would supplement the current understanding about burnout among HPs, and help inform interventions and strategies that mitigate the impact of burnout on HPs and the health system.

## Materials and methods

This study employed a cross-sectional, open and online survey to be completed voluntarily by HPs registered in Macao. The survey research design allowed data collection to be completed in a relatively short period so that a snapshot of the burnout phenomenon among HPs during the non-acute phase of the COVID-19 pandemic could be depicted in a timely manner amid the continuous progress of the situation. The survey was only open for 4 weeks around the second wave of COVID-19 infection cases (between 11 October and 8 November 2021). The reporting of this study was in compliance with the Strengthening the Reporting of Observational Studies in Epidemiology (STROBE) guideline and the Checklist for Reporting Results of Internet E-Surveys (CHERRIES) [36, 37].

### Study target

The target population of this study was all the HPs practicing in Macao such as doctors, nurses, pharmacists, medical laboratory technologists, Traditional Chinese Medicine doctors, etc. According to the official statistics, there were 10,566 HPs registered with the Health Bureau in Macau as of 2020. The valid sample size for the current study was determined at a minimum of 371 (confidence level 95%, margin of error 5%).

### Questionnaire design

The questionnaire used in this study consisted of 4 sections and was informed with previous literature and validated survey tools as shown in Table 1.

**Choice of the Copenhagen Burnout Inventory (CBI) to measure burnout.** The CBI was preferred in this study mainly because it: (1) specifically conceptualized burnout as a fatigue phenomenon; (2) distinguished burnout sourced from personal, work, and client factors, which was found particularly relevant to the purpose of this study; (3) focused on more on the "source" of burnout as opposed to symptoms, which fitted more adequately with the study purpose; (4) had been shown to have high internal reliability and validity [47]; and (5) has been used in recent studies about burnout among healthcare workers during the COVID-19 pandemic [48, 49]. By using the same tools, direct comparison across different studies may be more feasible. It is acknowledged that many previous studies on burnout among HPs predominately used a version of the Maslach Burnout Inventory (MBI) [50]. However, the MBI is only commercially available [51], and was designed to assess respondents' experiences of emotional exhaustion, depersonalization, and reduced personal accomplishment [52] which had been

**Table 1. The design of the questionnaire adopted in this study.**

| | |
|---|---|
| Section 1 | ■ The respondents were first asked to confirm if they were HPs practicing in Macao.<br>■ They were then asked to answer 6 questions about their demographic information (such as age [38], gender [17], marital status, parental status [39], highest education level [40], history of chronic diseases or severe illnesses [41]) and 7 questions about their professional background (such as type of profession [38], years of practice [42], level of seniority [38], management duty [42], place of practice [38], and their professional experiences with COVID-19 [38]). |
| Section 2 | ■ The Hospital Anxiety and Depression Scale (HADS) was used to screen the respondents for depression and anxiety [43].<br>■ There are 14 items in the HADS. Seven of the items relate to anxiety and seven relate to depression. There are 4 responses to each item, and each response is associated with a score of 0, 1, 2 or 3.<br>■ The possible total scores for anxiety and depression both range from 0 to 21. A total score between 0 and 7 was considered normal cases, a total score of 8–11 identified borderline cases and a total score of 12–21 indicated abnormal cases of anxiety or depression [44].<br>■ In this study, the Cronbach alpha reliability coefficients of the HADS were reasonable ($\alpha = 0.641$ and 0.647 for anxiety and depression respectively). |
| Section 3 | ■ The Copenhagen Burnout Inventory (CBI) was used to assess respondents' levels of burnout [45].<br>■ There are 3 subscales to the CBI: personal burnout (6 items), work burnout (7 items), and client burnout (6 items).<br>■ For each of the subscales, the total score is the average of the scores of the corresponding items. Subscale scores below 50 are considered "no/low burnout", scores of 50 to 74 are considered 'moderate burnout', scores of 75–99 are considered "high burnout", and a score of 100 is considered "severe burnout".<br>■ In this study, Cronbach alpha reliability coefficients of the CBI subscales were high (personal burnout $\alpha = 0.935$; work-related burnout $\alpha = 0.821$; and client-related burnout $\alpha = 0.900$). |
| Section 4 | ■ Respondents who had participated in the COVID-19 duties were invited to complete the questions in Section 4 which was designed to explore HP's perception of health workforce management during the COVID-19 pandemic and to identify areas which might help to mitigate burnout if improvements were made.<br>■ There were 14 items in this section, all of which were informed by the "Health workforce policy and management in the context of the COVID-19 pandemic response Interim Guidance" published by the WHO in December 2020 [46].<br>■ Respondents were asked to rate their level of agreement based on their professional experiences with COVID-19 related duties using a 5-point Likert scale with possible answers being strongly disagree, disagree, neutral, agree, and strongly agree. |

argued as being some of burnout symptoms [53]. In light of the above, the MBI was not employed for the purpose of this study.

**Health workforce management recommendations made by the WHO.** Section 4 of the questionnaire was informed with the "Health workforce policy and management in the context of the COVID-19 pandemic response Interim Guidance" produced by the WHO [46]. This guide aims to provide human resources recommendations for health managers and policy-makers to manage the COVID-19 pandemic covering 4 main domains: Domain 1—supporting and protecting health workers; Domain 2—strengthening and optimizing health workforce teams; Domain 3—increasing capacity and strategic health worker deployment; and Domain 4—health system human resources strengthening. Considering the target sample and the purpose of this study, which was to gain perspectives from frontline HPs, only Domain 1 (infection prevention and control, working conditions, mental health and psychosocial support, and remuneration) and Domain 2 (e.g. education and training, optimizing roles, leveraging community-based health workers) were used to inform the statements in Section 4 of the survey, while Domain 3 and Domain 4 were not as they related to higher-level decision-making.

## Development of the questionnaire

The questionnaire was designed and worded specifically to the context of HPs in Macao and was bilingual (English and Chinese) in order to minimize sampling bias due to language

barrier. Two rounds of pilot studies were conducted to ensure the validity. The initial instrument was first assessed by 3 researchers and 3 HPs, who were experienced in quantitative studies and public health measures, and fluent in both languages, through a focus group. Based on their feedback, adjustments to the wordings were made to improve clarity. The revised instrument was then pilot tested again on a convenience sample of 12 HPs (3 doctors, 3 nurses, 3 pharmacists, and 3 pharmacy technicians). They all agreed that the items were straight forward confirming the face and content validity of the questionnaire and found the online survey platform operated smoothly. No removal of the original items or addition of new ones was needed.

### Data collection

Convenience sampling was used to recruit participants and invitations were sent to as many HPs in Macao as possible through the membership networks of 5 major local healthcare professional organizations, social media like Facebook and WeChat, as well as the authors' professional networks. Snowball sampling technique was also used for enhancing recruitment.

It was clearly indicated in the Participant Information Statement (PIS) provided in the invitation and at the beginning of the questionnaire that, once the respondents had completed and submitted the survey online, they would be assumed to have given their consent to participate in this study. Moreover, the respondents were asked to check the box to indicate their consent before answering the survey questions. In the PIS, information about measures protecting participants' confidentiality was provided. No incentives were offered to the respondents upon completion of the survey.

The online survey was hosted by the online questionnaire distribution company "Survey Monkey" and was estimated to take around 6–8 minutes to complete. To ensure the completeness of the answers, a logic function requiring an answer to every question before submission was adopted. Only one attempt per device was allowed to avoid double entries or duplication of entries. Respondents were able to review and change their answers before submission, and pause and continue answering the online survey as long as the survey remained open. The number of people who visited the survey link was also recorded automatically.

### Data analysis

The survey responses were analyzed using the Statistical Package for Social Sciences (SPSS) version 27 software for Windows. Only the completed questionnaires were included for data analysis. In addition to descriptive statistics (frequencies, means, and standard deviations), the Pearson chi-square test was used to compare the differences in the level of burnout among subgroups, and Spearman's rho was used to test the correlation of burnout with respondents' experiences with COVID-19 related duties. Multiple logistic regressions for personal, work-related, and patient-related burnout were conducted separately in which moderate to severe burnout was set as dependent variable, and possible risk factors (demographic and professional characteristics, abnormal cases and borderline cases of depression and anxiety) as independent variables. Whenever the p-value is found to be smaller than 0.05, the association would be considered statistically significant at a confidence level of 95%.

## Results

As recorded by the online survey platform, a total of 622 people visited the survey link, of which 616 gave consent to participate, giving a participation rate of 99%. Out of the 616 respondents who agreed to participate and attempted the survey, 498 surveys were completed, giving a completion rate of 80.8%. As this was an anonymous survey, it was not able to find

out the reasons for not completing the survey from the respondents who did not do so. After the online survey was closed, a random follow-up contact with 24 HPs (5 doctors, 12 nurses, and 7 pharmacists) found that the main reasons they learnt about for respondents not completing the survey were a lack of time and the length of the survey. The average time taken to complete the survey was 8 minutes 53 seconds.

All respondents confirmed that their place of practice was in Macao. As shown in Table 2, the majority of the respondents aged between 25–44 years (n = 329, 66.1%), were female (n = 329, 66.1%), acquired a Bachelor degree as their highest education level (n = 309, 62.0%), and did not have any history of chronic or severe diseases (n = 399, 80.1%). Most of the respondents were either doctors (n = 187, 37.6%) or nurses (n = 135, 27.1%) along with other HPs such as medical laboratory technologists (n = 57, 11.4%) and pharmacy professionals (n = 54, 10.8%). A significant portion of them had less than 10 years of practice experiences (n = 229, n = 46%) or ranked themselves as junior staff (n = 200, 40.2%), while most indicated at least some management duties (n = 344, 69.1%). Healthcare providers from the public sector (n = 244, 49.0%) and the non-public sector (n = 254, 51.0%), and from hospital settings (n = 294, 59.0%) and non-hospital settings (n = 204, 41.0%) were reasonably represented. Most of the respondents had professional experiences with COVID-19 related duties (n = 413, 82.9%).

## Burnout

Moderate to severe levels of the 3 dimensions of burnout were found: 60.4% for personal burnout, 50.6% for work-related burnout, and 31.5% for patient-related burnout (Table 3). Regarding personal burnout, 43.2% of the respondents scored a moderate level, while 4.8% scored a severe level. One demographic attribute (age) and 2 professional attributes (year of practice, and working in the public sector) were found to be factors significantly associated with personal burnout. As for work-related burnout, 40.4% of the respondents scored moderate level while 0.4% scored severe level. Six demographic attributes (age, gender, marital status, parental status, highest education level, history of chronic or severe illnesses) and 4 professional attributes (year of practice, management duty, working in the hospital setting, involvement in COVID-19 related duties) were found to be factors significantly associated with work-related burnout. With client-related burnout, 24.9% of the respondents scored moderate level while 1.0% scored severe level. Four demographic attributes (age, marital status, parental status, and highest education level) and 4 professional attributes (year of practice, management duty, working in the hospital setting, and involvement in COVID-19 related duties) were found to be factors significantly associated with client-related burnout.

## Anxiety and depression

The normal, borderline, and abnormal cases of depression and anxiety among the respondents are shown in Table 4. Using the HADS, 39.1% (n = 195) and 30.1% (n = 150) of the respondents were identified as borderline cases of anxiety or depression, respectively; and 21.5% (n = 107) and 33.3% (n = 166) of the respondents were identified as abnormal cases of anxiety or depression, respectively. Both anxiety and depression were found to be significantly associated with all 3 dimensions of burnout (personal, work-related, and client-related).

## Logistic regression analysis

An analysis of standard residual (SR) showed that the data about burnout contained no outliers as none of the SRs were beyond ±3 (SR minimum = -2.435, SR maximum = 2.215 for personal burnout; SR minimum = -2.241, SR maximum = 2.087 for work-related burnout; SR

**Table 2. Descriptive information about the respondents (n = 498).**

| Demographic characteristics | | Cases | |
|---|---|---|---|
| | | n | % |
| **Age (years)** | 18–24 | 20 | 4.0 |
| | 25–34 | 176 | 35.3 |
| | 35–44 | 153 | 30.7 |
| | 45–54 | 84 | 16.9 |
| | 55–64 | 54 | 10.8 |
| | 65+ | 11 | 2.2 |
| **Gender** | Female | 329 | 66.1 |
| | Male | 157 | 31.5 |
| | Prefer not to indicate | 12 | 2.4 |
| **Marital status** | Married | 283 | 56.8 |
| | Single[1] | 190 | 38.2 |
| | Prefer not to indicate | 25 | 5.0 |
| **Being a parent** | No | 232 | 46.6 |
| | Yes | 266 | 53.4 |
| **Highest Education level** | Bachelor | 309 | 62.0 |
| | Master | 156 | 31.3 |
| | PhD | 23 | 4.6 |
| | Other | 10 | 2.0 |
| **History of chronic diseases or severe illnesses** | No | 399 | 80.1 |
| | Yes | 99 | 19.9 |
| **Professional category** | Doctor | 187 | 37.6 |
| | Dentist | 26 | 5.2 |
| | TCM doctor | 39 | 7.8 |
| | Pharmacist | 33 | 6.6 |
| | Nurse | 135 | 27.1 |
| | Medical laboratory technologist | 57 | 11.4 |
| | Pharmacy technician | 21 | 4.2 |
| **Years of practice (years)** | ≤10 | 229 | 46.0 |
| | 11–20 | 129 | 25.9 |
| | 21–30 | 87 | 17.5 |
| | 30+ | 53 | 10.6 |
| **Level of seniority** | Intern | 6 | 1.2 |
| | Resident/Junior | 200 | 40.2 |
| | Specialist/Superior | 145 | 29.1 |
| | Consultant | 35 | 7.0 |
| | Non-practicing | 5 | 1.0 |
| | Other | 107 | 21.5 |
| **Management duty** | No (absence of staff from the lower levels) | 154 | 30.9 |
| | Yes (presence of staff from the lower levels) | 344 | 69.1 |
| **Working in the public sector** | Non-public sector[2] | 254 | 51.0 |
| | Public sector[3] | 244 | 49.0 |
| **Working in a hospital setting** | Non-hospital setting | 204 | 41.0 |
| | Hospital setting | 294 | 59.0 |

(*Continued*)

**Table 2.** (Continued)

| Demographic characteristics | | Cases | |
|---|---|---|---|
| | | **n** | **%** |
| **Professional experiences with COVID-19 related duties** | No | **85** | 17.1 |
| | Yes[4] | **413** | 82.9 |

1—Single status including never married, separated, widowed, and divorced; 2. Public sector including public hospital (n = 186, 37.3%), primary health center (n = 58, 11.6%); 3. Private sector including non-public hospital (n = 108, 21.7%), private clinic (n = 91, 18.3%), non-government organization (n = 14, 2.8%), community pharmacy (n = 15, 3.0%), others (n = 26, 5.2%); 4—COVID-19 related professional experiences including direct care of COVID-19 patients (n = 33, 6.6%), COVID-19 related back office duties (n = 145, 29.1%), regular COVID-19 testing (n = 132, 26.5%), mass COVID-19 testing (n = 298, 59.8%), COVID-19 vaccination (n = 113, 22.7%), volunteer work related to COVID-19 (n = 77, 15.5%), other additional duties due to COVID-19 (n = 38, 7.6%)

minimum = -1.75, SR maximum = 2.214 for client-related burnout). Tests were also conducted to show multicollinearity was not a concern for the 15 variables (including demographic and professional characteristics, as well as anxiety and depression status) considering that tolerance for all the variables was greater than 0.1 and none of the variance inflation factor (VIF) was greater than 5.

Logistic regression analysis showed that 5, 6 and 4 variables were respectively associated with personal, work-related, and client-related burnout with a statistical difference as shown in Table 5. The unadjusted odds ratios indicated that: (1) HPs in the public sector were 2.088 (95% CI 1.160–3.758) times more likely to experience moderate to severe personal burnout compared to HPs in non-public sector; (2) HPs in the hospital setting were 2.280 (95% CI 1.319–3.943) times more likely to experience moderate to severe work-related burnout compared to HPs in non-hospital setting; and (3) HPs who undertook COVID-19 professional duties were 2.421 (95% CI 1.218–4.811) times more likely to experience moderate to severe client-related burnout compared to HPs who did not.

In addition, HPs with abnormal cases of anxiety were 14.326 times (95% CI 4.317–47.543), 8.169 times (95% CI 3.662–18.224) and 3.211 times (95% CI 1.525–6.762) more like to experience moderate to severe personal, work-related and client-related burnout respectively when compared to HPs with no anxiety. Similarly, HPs with abnormal cases of depression were 12.050 times (95% CI 5.420–26.791), 3.965 times (95% CI 2.034–7.729) and 4.180 times (95% CI 2.074–8.423) more like to experience moderate to severe personal, work-related and client-related burnout respectively when compared to HPs with no depression.

## HPs' experiences with COVID-19 related duties

Among 413 respondents who had participated in the COVID-19 related duties, 396 opted to continue into Section 4 of the questionnaire and rated the 14 statements about their experiences with COVID-19 related duties (Table 6). HPs participated in COVID-19 duties were mostly dissatisfied with the accessibility of psychological support at workplace (62.6%), workforce distribution for COVID-19 duties (50.0%), ability to rest and recover (46.2%), and remuneration (44.7%). It is also worth noting that 36.4% of the respondents felt that they were at risk of being infected with COVID-19 at their workplace, and 33.1% did not think their superior was supportive of their participation in COVD-19 duties.

Only about half of the respondents believed that they had enough knowledge about COVID-19 to manage the duties (52.3%). Less than 40% thought they were healthy enough to participate in COVID-19 related duties and 18.7% indicated a need for psychological support.

**Table 3. Levels of burnout among the respondents (n = 498).**

| | Cases (n) | Personal Burnout | | | | | | Work-related burnout | | | | | | Client-related burnout | | | | | |
|---|---|---|---|---|---|---|---|---|---|---|---|---|---|---|---|---|---|---|---|
| | | No/Low (n = 197, 39.6%) % | Moderate (n = 215, 43.2%) % | High (n = 62, 12.4%) % | Severe (n = 24, 4.8%) % | X2 | p | No/Low (n = 246, 49.4%) % | Moderate (n = 201, 40.4%) % | High (n = 49, 9.8%) % | Severe (n = 2, 0.4%) % | X2 | p | No/Low (n = 341, 68.5%) % | Moderate (n = 124, 24.9%) % | High (n = 28, 5.6%) % | Severe (n = 5, 1.0%) % | X2 | p |
| **Age (years)** | | | | | | | | | | | | | | | | | | | |
| 18–24 | 20 | 40.0% | 35.0% | 20.0% | 5.0% | -0.173** | 0.000 | 40.0% | 40.0% | 20.0% | 0.0% | -0.250** | 0.000 | 70.0% | 25.0% | 5.0% | 0.0% | -0.206** | 0.000 |
| 25–34 | 176 | 30.1% | 48.9% | 13.6% | 7.4% | | | 35.2% | 50.0% | 13.6% | 1.1% | | | 58.0% | 29.5% | 10.2% | 2.3% | | |
| 35–44 | 153 | 39.9% | 41.2% | 15.7% | 3.3% | | | 51.6% | 40.5% | 7.8% | 0.0% | | | 68.6% | 26.8% | 4.6% | 0.0% | | |
| 45–54 | 84 | 47.6% | 40.5% | 7.1% | 4.8% | | | 60.7% | 31.0% | 8.3% | 0.0% | | | 75.0% | 21.4% | 2.4% | 1.2% | | |
| 55–64 | 54 | 51.9% | 40.7% | 5.6% | 1.9% | | | 66.7% | 29.6% | 3.7% | 0.0% | | | 87.0% | 13.0% | 0.0% | 0.0% | | |
| 65+ | 11 | 63.6% | 27.3% | 9.1% | 0.0% | | | 90.9% | 9.1% | 0.0% | 0.0% | | | 90.9% | 9.1% | 0.0% | 0.0% | | |
| **Gender** | | | | | | | | | | | | | | | | | | | |
| Female | 329 | 38.0% | 45.6% | 12.2% | 4.3% | 0.026 | 0.565 | 49.8% | 42.2% | 7.6% | 0.3% | 0.091* | 0.043 | 69.9% | 24.6% | 4.6% | 0.9% | 0.084 | 0.062 |
| Male | 157 | 43.3% | 40.1% | 11.5% | 5.1% | | | 51.0% | 35.7% | 12.7% | 0.6% | | | 67.5% | 24.8% | 6.4% | 1.3% | | |
| Prefer not to indicate | 12 | 33.3% | 16.7% | 33.3% | 16.7% | | | 16.7% | 50.0% | 33.3% | 0.0% | | | 41.7% | 33.3% | 25.0% | 0.0% | | |
| **Marital status** | | | | | | | | | | | | | | | | | | | |
| Married | 283 | 41.0% | 42.4% | 14.1% | 2.5% | 0.037 | 0.414 | 53.7% | 40.3% | 6.0% | 0.0% | 0.125** | 0.005 | 72.4% | 24.4% | 2.8% | 0.4% | 0.111* | 0.013 |
| Single | 190 | 34.7% | 47.9% | 8.9% | 8.4% | | | 42.1% | 42.1% | 14.7% | 1.1% | | | 62.1% | 25.8% | 10.0% | 2.1% | | |
| Prefer not to indicate | 25 | 60.0% | 16.0% | 20.0% | 4.0% | | | 56.0% | 28.0% | 16.0% | 0.0% | | | 72.0% | 24.0% | 4.0% | 0.0% | | |
| **Being a parent** | | | | | | | | | | | | | | | | | | | |
| No | 232 | 37.5% | 44.4% | 11.2% | 6.9% | -0.056 | 0.211 | 44.4% | 40.9% | 13.8% | 0.9% | -0.136** | 0.002 | 64.2% | 24.6% | 9.5% | 1.7% | -0.139** | 0.002 |
| Yes | 266 | 41.4% | 42.1% | 13.5% | 3.0% | | | 53.8% | 39.8% | 6.4% | 0.0% | | | 72.2% | 25.2% | 2.3% | 0.4% | | |
| **Highest Education level** | | | | | | | | | | | | | | | | | | | |
| Bachelor | 309 | 37.9% | 44.0% | 12.9% | 5.2% | -0.048 | 0.283 | 45.6% | 41.7% | 12.3% | 0.3% | -0.092* | 0.040 | 64.4% | 28.5% | 5.8% | 1.3% | -0.110* | 0.014 |
| Master | 156 | 40.4% | 45.5% | 10.9% | 3.2% | | | 55.8% | 39.7% | 3.8% | 0.6% | | | 73.7% | 20.5% | 5.1% | 0.6% | | |
| PhD | 23 | 43.5% | 26.1% | 17.4% | 13.0% | | | 47.8% | 34.8% | 17.4% | 0.0% | | | 73.9% | 17.4% | 8.7% | 0.0% | | |
| Other | 10 | 70.0% | 20.0% | 10.0% | 0.0% | | | 70.0% | 20.0% | 10.0% | 0.0% | | | ###### | 0.0% | 0.0% | 0.0% | | |
| **Chronic diseases or severe illnesses** | | | | | | | | | | | | | | | | | | | |
| Yes | 399 | 40.4% | 42.4% | 13.1% | 4.0% | 0.010 | 0.817 | 59.6% | 33.3% | 6.1% | 1.0% | 0.094* | 0.036 | 72.7% | 22.2% | 5.1% | 0.0% | 0.053 | 0.237 |
| No | 99 | 39.3% | 43.4% | 12.3% | 5.0% | | | 46.9% | 42.1% | 10.8% | 0.3% | | | 67.4% | 25.6% | 5.8% | 1.3% | | |
| **Professional category** | | | | | | | | | | | | | | | | | | | |
| Doctor | 187 | 43.9% | 38.5% | 12.3% | 5.3% | 0.048 | 0.282 | 52.9% | 37.4% | 8.6% | 1.1% | 0.062 | 0.168 | 67.9% | 25.7% | 5.3% | 1.1% | 0.024 | 0.595 |
| Dentist | 26 | 38.5% | 50.0% | 7.7% | 3.8% | | | 46.2% | 46.2% | 7.7% | 0.0% | | | 73.1% | 19.2% | 7.7% | 0.0% | | |
| TCM doctor | 39 | 46.2% | 41.0% | 12.8% | 0.0% | | | 59.0% | 38.5% | 2.6% | 0.0% | | | 71.8% | 23.1% | 5.1% | 0.0% | | |
| Pharmacist | 33 | 54.5% | 33.3% | 9.1% | 3.0% | | | 48.5% | 48.5% | 3.0% | 0.0% | | | 78.8% | 15.2% | 6.1% | 0.0% | | |
| Nurse | 135 | 27.4% | 51.9% | 14.1% | 6.7% | | | 38.5% | 45.9% | 15.6% | 0.0% | | | 63.0% | 28.1% | 7.4% | 1.5% | | |
| Medical laboratory technologist | 57 | 38.6% | 45.6% | 14.0% | 1.8% | | | 59.6% | 33.3% | 7.0% | 0.0% | | | 75.4% | 21.1% | 3.5% | 0.0% | | |
| Pharmacy technician | 21 | 47.6% | 33.3% | 9.5% | 9.5% | | | 47.6% | 33.3% | 19.0% | 0.0% | | | 61.9% | 33.3% | 0.0% | 4.8% | | |
| **Years of practice** | | | | | | | | | | | | | | | | | | | |
| <10 | 229 | 33.6% | 46.3% | 13.5% | 6.6% | -0.159** | 0.000 | 38.4% | 47.2% | 13.5% | 0.9% | -0.222** | 0.000 | 60.3% | 27.5% | 10.5% | 1.7% | -0.219** | 0.000 |

(Continued)

**Table 3.** (Continued)

| | Cases (n) | Personal Burnout | | | | | | Work-related burnout | | | | | | Client-related burnout | | | | | |
|---|---|---|---|---|---|---|---|---|---|---|---|---|---|---|---|---|---|---|---|
| | | No/Low (n = 197, 39.6%) % | Moderate (n = 215, 43.2%) % | High (n = 62, 12.4%) % | Severe (n = 24, 4.8%) % | X2 | p | No/Low (n = 246, 49.4%) % | Moderate (n = 201, 40.4%) % | High (n = 49, 9.8%) % | Severe (n = 2, 0.4%) % | X2 | p | No/Low (n = 341, 68.5%) % | Moderate (n = 124, 24.9%) % | High (n = 28, 5.6%) % | Severe (n = 5, 1.0%) % | X2 | p |
| 11–20 | 129 | 37.2% | 41.1% | 17.1% | 4.7% | | | 52.7% | 40.3% | 7.0% | 0.0% | | | 69.8% | 27.1% | 3.1% | 0.0% | | |
| 21–30 | 87 | 55.2% | 35.6% | 5.7% | 3.4% | | | 60.9% | 31.0% | 8.0% | 0.0% | | | 78.2% | 20.7% | 0.0% | 1.1% | | |
| 30+ | 53 | 45.3% | 47.2% | 7.5% | 0.0% | | | 69.8% | 26.4% | 3.8% | 0.0% | | | 84.9% | 15.1% | 0.0% | 0.0% | | |
| **Professional seniority** | | | | | | −0.038 | 0.400 | | | | | −0.079 | 0.078 | | | | | −0.063 | 0.158 |
| Intern | 6 | 50.0% | 33.3% | 16.7% | 0.0% | | | 50.0% | 33.3% | 16.7% | 0.0% | | | 66.7% | 16.7% | 16.7% | 0.0% | | |
| Resident/Junior | 200 | 35.0% | 46.5% | 13.0% | 5.5% | | | 43.0% | 44.5% | 11.5% | 1.0% | | | 63.5% | 27.0% | 7.5% | 2.0% | | |
| Specialist/Superior | 145 | 44.1% | 37.2% | 13.1% | 5.5% | | | 52.4% | 38.6% | 9.0% | 0.0% | | | 71.7% | 24.8% | 3.4% | 0.0% | | |
| Consultant | 35 | 40.0% | 54.3% | 2.9% | 2.9% | | | 65.7% | 25.7% | 8.6% | 0.0% | | | 82.9% | 14.3% | 0.0% | 2.9% | | |
| Non-practicing | 5 | 60.0% | 40.0% | 0.0% | 0.0% | | | 60.0% | 40.0% | 0.0% | 0.0% | | | 60.0% | 20.0% | 20.0% | 0.0% | | |
| Other | 107 | 40.2% | 42.1% | 14.0% | 3.7% | | | 51.4% | 40.2% | 8.4% | 0.0% | | | 69.2% | 25.2% | 5.6% | 0.0% | | |
| **Management duty** | | | | | | 0.037 | 0.406 | | | | | 0.118** | 0.009 | | | | | 0.151** | 0.001 |
| Yes | 154 | 42.9% | 40.9% | 11.7% | 4.5% | | | 57.8% | 35.1% | 7.1% | 0.0% | | | 77.9% | 19.5% | 2.6% | 0.0% | | |
| No | 344 | 38.1% | 44.2% | 12.8% | 4.9% | | | 45.6% | 42.7% | 11.0% | 0.6% | | | 64.2% | 27.3% | 7.0% | 1.5% | | |
| **Working in the public sector** | | | | | | 0.125** | 0.005 | | | | | 0.051 | 0.258 | | | | | 0.066 | 0.144 |
| Non-public sector | 254 | 46.9% | 38.2% | 10.6% | 4.3% | | | 50.4% | 41.3% | 8.3% | 0.0% | | | 71.3% | 23.2% | 4.7% | 0.8% | | |
| Public sector | 244 | 32.0% | 48.4% | 14.3% | 5.3% | | | 48.4% | 39.3% | 11.5% | 0.8% | | | 65.6% | 26.6% | 6.6% | 1.2% | | |
| **Working in a hospital setting** | | | | | | 0.086 | 0.056 | | | | | 0.175** | 0.000 | | | | | 0.095* | 0.034 |
| Non-hospital setting | 204 | 44.6% | 39.7% | 12.7% | 2.9% | | | 58.3% | 36.3% | 5.4% | 0.0% | | | 72.1% | 24.5% | 2.9% | 0.5% | | |
| Hospital setting | 294 | 36.1% | 45.6% | 12.2% | 6.1% | | | 43.2% | 43.2% | 12.9% | 0.7% | | | 66.0% | 25.2% | 7.5% | 1.4% | | |
| **Professional experience with COVID-19** | | | | | | 0.072 | 0.108 | | | | | 0.095* | 0.034 | | | | | 0.119** | 0.008 |
| No | 85 | 49.4% | 34.1% | 14.1% | 2.4% | | | 60.0% | 32.9% | 7.1% | 0.0% | | | 81.2% | 15.3% | 3.5% | 0.0% | | |
| Yes | 413 | 37.5% | 45.0% | 12.1% | 5.3% | | | 47.2% | 41.9% | 10.4% | 0.5% | | | 65.9% | 26.9% | 6.1% | 1.2% | | |

\* p<0.05

\*\* p<0.01

**Table 4. Prevalence of burnout, depression and anxiety among the participants (n = 498).**

| | | Personal Burnout | | | | | | | | $X^2$ | $p$ |
|---|---|---|---|---|---|---|---|---|---|---|---|
| | | No/Low (n = 197) | | Moderate (n = 215) | | High (n = 62) | | Severe (n = 24) | | | |
| | | n | % | n | % | n | % | n | % | | |
| **Anxiety** | Normal (n = 196, 39.4%) | 127 | 64.8% | 62 | 31.6% | 6 | 3.1% | 1 | 0.5% | 0.549** | 0.000 |
| | Borderline (n = 195, 39.1%) | 66 | 33.8% | 102 | 52.3% | 24 | 12.3% | 3 | 1.5% | | |
| | Abnormal (n = 107, 21.5%) | 4 | 3.7% | 51 | 47.7% | 32 | 29.9% | 20 | 18.7% | | |
| **Depression** | Normal (n = 182, 36.6%) | 124 | 68.1% | 50 | 27.5% | 5 | 2.7% | 3 | 1.6% | 0.481** | 0.000 |
| | Borderline (n = 150, 30.1%) | 55 | 36.7% | 73 | 48.7% | 18 | 12.0% | 4 | 2.7% | | |
| | Abnormal (n = 166, 33.3%) | 18 | 10.8% | 92 | 55.4% | 39 | 23.5% | 17 | 10.2% | | |
| | | Work-related burnout | | | | | | | | $X^2$ | $P$ |
| | | No/Low (n = 246) | | Moderate (n = 201) | | High (n = 49) | | Severe (n = 2) | | | |
| | | n | % | n | % | n | % | n | % | | |
| **Anxiety** | Normal (n = 196, 39.4%) | 144 | 73.5% | 48 | 24.5% | 4 | 2.0% | 0 | 0.0% | 0.494** | 0.000 |
| | Borderline (n = 195, 39.1%) | 86 | 44.1% | 97 | 49.7% | 12 | 6.2% | 0 | 0.0% | | |
| | Abnormal (n = 107, 21.5%) | 16 | 15.0% | 56 | 52.3% | 33 | 30.8% | 2 | 1.9% | | |
| **Depression** | Normal (n = 182, 36.6%) | 131 | 72.0% | 42 | 23.1% | 9 | 4.9% | 0 | 0.0% | 0.376** | 0.000 |
| | Borderline (n = 150, 30.1%) | 70 | 46.7% | 72 | 48.0% | 7 | 4.7% | 1 | 0.7% | | |
| | Abnormal (n = 166, 33.3%) | 45 | 27.1% | 87 | 52.4% | 33 | 19.9% | 1 | 0.6% | | |
| | | Client-related burnout | | | | | | | | $X^2$ | $P$ |
| | | No/Low (n = 341) | | Moderate (n = 124) | | High (n = 28) | | Severe (n = 5) | | | |
| | | n | % | n | % | n | % | n | % | | |
| **Anxiety** | Normal (n = 196, 39.4%) | 166 | 84.7% | 28 | 14.3% | 2 | 1.0% | 0 | 0.0% | 0.379** | 0.000 |
| | Borderline (n = 195, 39.1%) | 131 | 67.2% | 56 | 28.7% | 6 | 3.1% | 2 | 1.0% | | |
| | Abnormal (n = 107, 21.5%) | 44 | 41.1% | 40 | 37.4% | 20 | 18.7% | 3 | 2.8% | | |
| **Depression** | Normal (n = 182, 36.6%) | 156 | 85.7% | 22 | 12.1% | 3 | 1.6% | 1 | 0.5% | 0.308** | 0.000 |
| | Borderline (n = 150, 30.1%) | 103 | 68.7% | 38 | 25.3% | 8 | 5.3% | 1 | 0.7% | | |
| | Abnormal (n = 166, 33.3%) | 82 | 49.4% | 64 | 38.6% | 17 | 10.2% | 3 | 1.8% | | |

\* $p < 0.05$

\*\* $p < 0.01$

About 1 in 2 HPs experienced a sense of job satisfaction when participating in COVID-19 duties (48.0%). The ratings of all 14 statements were found to be significantly associated with moderate to severe personal, work-related, and client-related burnout.

Spearman's correlation was computed to assess the relationship between the level of agreements on the experiences with COVID-19 related duties to the likelihood of moderate to severe personal, work-related, and client-related burnout. Apart from statements 1 and 3, all the other statements were negatively associated with burnout. The positive values of Spearman's rho in Statements 1 (r(df) = 0.275, $p < 0.01$; r(df) = 0.259, $p<0.01$; r(df) = 0.257, $p<0.01$) and Statement 3 (r(df) = 0.145, $p<0.05$; r(df) = 0.133, $p<0.05$; r(df) = 0.201, $p<0.01$) indicated that the more the respondents perceived a risk of infection with COVID-19 at workplace or had the need for psychological support, the more likely they were going to experience different types of moderate to severe burnout.

On the contrary, the negative values of Spearman's rho in all the remaining statements which described different sources of support, sound management practice and perceived health status indicated that all these factors were negatively associated with the likelihood of

**Table 5. Logistic regression analysis of factors associated with moderate to severe burnout.**

| Variables | | Personal burnout | | | | Work-related burnout | | | | Client-related burnout | | | |
|---|---|---|---|---|---|---|---|---|---|---|---|---|---|
| | | OR | P value | 95% CI | | OR | P value | 95% CI | | OR | P value | 95% CI | |
| | | | | Lower | Upper | | | Lower | Upper | | | Lower | Upper |
| **Age (years)** | 18–24 | | | | | | | | | | | | |
| | 25–34 | 2.757 | 0.140 | 0.716 | 10.613 | 1.985 | 0.283 | 0.567 | 6.951 | 1.771 | 0.324 | 0.569 | 5.512 |
| | 35–44 | 1.338 | 0.711 | 0.287 | 6.246 | 1.312 | 0.715 | 0.306 | 5.628 | 1.413 | 0.615 | 0.368 | 5.421 |
| | 45–54 | 0.856 | 0.865 | 0.142 | 5.166 | 0.809 | 0.808 | 0.146 | 4.489 | 1.091 | 0.917 | 0.213 | 5.575 |
| | 55–64 | 0.503 | 0.509 | 0.066 | 3.855 | 0.707 | 0.729 | 0.100 | 5.016 | 0.414 | 0.382 | 0.057 | 2.990 |
| | 65+ | 0.291 | 0.362 | 0.021 | 4.129 | 0.102 | 0.126 | 0.005 | 1.894 | 0.590 | 0.718 | 0.034 | 10.330 |
| **Gender** | Female | | | | | | | | | | | | |
| | Male | 0.903 | 0.713 | 0.524 | 1.555 | 0.981 | 0.939 | 0.592 | 1.625 | 1.296 | 0.287 | 0.804 | 2.091 |
| | Prefer not to indicate | 0.897 | 0.910 | 0.137 | 5.868 | 6.296 | 0.085 | 0.776 | 51.068 | 4.091 | 0.057 | 0.960 | 17.429 |
| **Marital status** | Married | | | | | | | | | | | | |
| | Single | 0.949 | 0.905 | 0.405 | 2.223 | 1.254 | 0.547 | 0.601 | 2.617 | 1.557 | 0.210 | 0.780 | 3.107 |
| | Prefer not to indicate | 0.632 | 0.500 | 0.167 | 2.396 | 0.812 | 0.762 | 0.211 | 3.128 | 0.984 | 0.980 | 0.281 | 3.450 |
| **Being a parent** | No | | | | | | | | | | | | |
| | Yes | 1.303 | 0.547 | 0.551 | 3.084 | 1.752 | 0.142 | 0.829 | 3.705 | 1.537 | 0.244 | 0.745 | 3.171 |
| **Highest Education level** | Bachelor | | | | | | | | | | | | |
| | Master | 1.231 | 0.463 | 0.707 | 2.144 | 0.760 | 0.299 | 0.453 | 1.275 | 0.652 | 0.083 | 0.403 | 1.057 |
| | PhD | 0.965 | 0.952 | 0.301 | 3.097 | 1.179 | 0.760 | 0.409 | 3.397 | 0.721 | 0.561 | 0.240 | 2.171 |
| **History of chronic diseases or severe illnesses** | No | | | | | | | | | | | | |
| | Yes | 1.305 | 0.426 | 0.677 | 2.516 | 0.772 | 0.403 | 0.422 | 1.415 | 1.046 | 0.879 | 0.587 | 1.864 |
| **Years of practice** | 10 years or less | | | | | | | | | | | | |
| | 11–20 years | 0.795 | 0.599 | 0.338 | 1.870 | 0.454 | 0.056 | 0.202 | 1.021 | 0.585 | 0.176 | 0.269 | 1.273 |
| | 21–30 years | 0.326 | 0.094 | 0.088 | 1.208 | 0.487 | 0.241 | 0.147 | 1.620 | 0.529 | 0.288 | 0.163 | 1.712 |
| | More than 30 years | 0.731 | 0.715 | 0.136 | 3.918 | 0.457 | 0.322 | 0.097 | 2.156 | 0.592 | 0.522 | 0.119 | 2.943 |
| **Management duty** | No | | | | | | | | | | | | |
| | Yes | 1.510 | 0.193 | 0.813 | 2.805 | 0.877 | 0.649 | 0.499 | 1.543 | 0.679 | 0.157 | 0.397 | 1.161 |
| **Working the public sector** | No | | | | | | | | | | | | |
| | Yes | 2.088 | *0.014** | 1.160 | 3.758 | 0.864 | 0.601 | 0.500 | 1.494 | 1.268 | 0.351 | 0.770 | 2.090 |
| **Work in the hospital setting** | No | | | | | | | | | | | | |
| | Yes | 1.222 | 0.505 | 0.678 | 2.202 | 2.280 | *0.003*** | 1.319 | 3.943 | 0.954 | 0.856 | 0.574 | 1.586 |
| **Professional experience with COVID-19 duties** | No | | | | | | | | | | | | |
| | Yes | 1.170 | 0.657 | 0.585 | 2.338 | 1.285 | 0.456 | 0.665 | 2.485 | 2.421 | *0.012** | 1.218 | 4.811 |
| **Anxiety** | Normal | | | | | | | | | | | | |
| | Borderline cases | 1.767 | *0.038** | 1.031 | 3.030 | 2.203 | *0.003*** | 1.300 | 3.732 | 1.444 | 0.231 | 0.792 | 2.633 |
| | Abnormal cases | 14.326 | *0.000*** | 4.317 | 47.543 | 8.169 | *0.000*** | 3.662 | 18.224 | 3.211 | *0.002*** | 1.525 | 6.762 |
| **Depression** | Normal | | | | | | | | | | | | |
| | Borderline cases | 3.102 | *0.000*** | 1.766 | 5.451 | 2.455 | *0.002*** | 1.396 | 4.315 | 2.531 | *0.004*** | 1.341 | 4.777 |
| | Abnormal cases | 12.050 | *0.000*** | 5.420 | 26.791 | 3.965 | *0.000*** | 2.034 | 7.729 | 4.180 | *0.000*** | 2.074 | 8.423 |

* p<0.05

** p<0.01

burnout. According to the results of Statement 14, high level of perceived health (r(df) = -0.488, $p < 0.01$; r(df) = -0.417, $p < 0.01$; r(df) = -0.411, $p < 0.01$) had the strongest negative association with moderate to severe burnout. Other important factors such as Statement 13—

Table 6. Respondents' experiences with COVID-19 related duties and the association with burnout (n = 396).

| Statements about the experiences of COVID-19 related duties | Mean | SD | Frequency | | | | | | | | | | Association with moderate to severe personal burnout | | Association with moderate to severe work-related burnout | | Association with moderate to severe client-related burnout | |
|---|---|---|---|---|---|---|---|---|---|---|---|---|---|---|---|---|---|---|
| | | | Highly disagree | | Disagree | | Neither agree or disagree | | Agree | | Highly agree | | Spearman's rho | P | Spearman's rho | P | Spearman's rho | P |
| | | | n | % | n | % | n | % | n | % | n | % | | | | | | |
| 1. "I feel at risk of being infected by COVID-19 at my workplace." | 2.79 | ±1.19 | 36 | 9.1% | 101 | 25.5% | 115 | 29.0% | 95 | 24.0% | 49 | 12.4% | 0.275** | 0.000 | 0.259** | 0.000 | 0.257** | 0.000 |
| 2. "I am provided with adequate supply of Personal Protective Equipment at my workplace." | 3.56 | ±1.25 | 20 | 5.1% | 47 | 11.9% | 73 | 18.4% | 120 | 30.3% | 136 | 34.3% | -0.140** | 0.005 | -0.206** | 0.000 | -0.213** | 0.000 |
| 3. "I have the need to receive psychological support." | 1.93 | ±1.38 | 165 | 41.7% | 75 | 18.9% | 82 | 20.7% | 43 | 10.9% | 31 | 7.8% | 0.145** | 0.004 | 0.133** | 0.008 | 0.201** | 0.000 |
| 4. "I am able to receive psychological support at my workplace." | 1.90 | ±1.36 | 169 | 42.7% | 79 | 19.9% | 74 | 18.7% | 47 | 11.9% | 27 | 6.8% | -0.238** | 0.000 | -0.251** | 0.000 | -0.205** | 0.000 |
| 5. "I am reasonably remunerated for the COVID-19 duties that I performed." | 2.41 | ±1.50 | 121 | 30.6% | 56 | 14.1% | 92 | 23.2% | 75 | 18.9% | 52 | 13.1% | -0.239** | 0.000 | -0.295** | 0.000 | -0.317** | 0.000 |
| 6. "I have enough COVID-19 knowledge to manage my duties at work." | 3.24 | ±1.14 | 17 | 4.3% | 67 | 16.9% | 105 | 26.5% | 128 | 32.3% | 79 | 19.9% | -0.173** | 0.001 | -0.274** | 0.000 | -0.206** | 0.000 |
| 7. "The health workforce distribution regarding COVID-19 duties is reasonable at my workplace." | 2.21 | ±1.20 | 101 | 25.5% | 97 | 24.5% | 111 | 28.0% | 69 | 17.4% | 18 | 4.5% | -0.291** | 0.000 | -0.398** | 0.000 | -0.329** | 0.000 |
| 8. "The COVID-19 duties are distributed fairly within the health workforce that I belong to." | 2.52 | ±1.28 | 79 | 19.9% | 81 | 20.5% | 116 | 29.3% | 88 | 22.2% | 32 | 8.1% | -0.286** | 0.000 | -0.345** | 0.000 | -0.335** | 0.000 |
| 9. "My superior is supportive of me when participating in COVID-19 related duties." | 3.01 | ±1.45 | 63 | 15.9% | 68 | 17.2% | 73 | 18.4% | 100 | 25.3% | 92 | 23.2% | -0.307** | 0.000 | -0.393** | 0.000 | -0.378** | 0.000 |

*(Continued)*

**Table 6.** (Continued)

| Statements about the experiences of COVID-19 related duties | Mean | SD | Frequency | | | | | | | | | | Association with moderate to severe personal burnout | | Association with moderate to severe work-related burnout | | Association with moderate to severe client-related burnout | |
|---|---|---|---|---|---|---|---|---|---|---|---|---|---|---|---|---|---|---|
| | | | Highly disagree | | Disagree | | Neither agree or disagree | | Agree | | Highly agree | | Spearman's rho | P | Spearman's rho | P | Spearman's rho | P |
| | | | n | % | n | % | n | % | n | % | n | % | | | | | | |
| 10. "My workload is manageable when participating in COVID-19 related duties." | 2.95 | ±1.20 | 37 | 9.3% | 79 | 19.9% | 107 | 27.0% | 121 | 30.6% | 52 | 13.1% | -0.362** | 0.000 | -0.410** | 0.000 | -0.371** | 0.000 |
| 11. "I experience a sense of job satisfaction when participating in COVID-19 related duties." | 3.02 | ±1.38 | 58 | 14.6% | 61 | 15.4% | 87 | 22.0% | 110 | 27.8% | 80 | 20.2% | -0.338** | 0.000 | -0.455** | 0.000 | -0.408** | 0.000 |
| 12. "I am able to take a break when participating in COVID-19 related duties." | 2.62 | ±1.20 | 62 | 15.7% | 84 | 21.2% | 121 | 30.6% | 98 | 24.7% | 31 | 7.8% | -0.325** | 0.000 | -0.327** | 0.000 | -0.320** | 0.000 |
| 13. "I am able to rest and recover during breaks when participating in COVID-19 related duties." | 2.41 | ±1.24 | 76 | 19.2% | 107 | 27.0% | 113 | 28.5% | 68 | 17.2% | 32 | 8.1% | -0.384** | 0.000 | -0.371** | 0.000 | -0.308** | 0.000 |
| 14. "I am healthy enough to participate COVID-19 related duties." | 2.86 | ±1.22 | 40 | 10.1% | 81 | 20.5% | 120 | 30.3% | 107 | 27.0% | 48 | 12.1% | -0.488** | 0.000 | -0.417** | 0.000 | -0411** | 0.000 |

* p<0.05

** p<0.01

being able to rest and recover during breaks when participating in COVID-19 related duties (r(df) = -0.384, *p<0.01*), Statement 11—a sense of job satisfaction when participating in COVID-19 related duties (r(df) = -0.455, *p<0.01*), and Statement 9 –supportive superior (r(df) = -0.378, *p<0.01*) also showed significant negative association with moderate to severe personal, work-related, and client-related burnout respectively.

## Discussion

To our knowledge, this is one of the very few studies that assessed burnout in HPs during a time of very low incidence of COVID-19 infections and explored HPs' experiences of anti-COVID-19 duties in order to inform health workforce management that might mitigate HPs' mental health concerns. This study found a significant burden of burnout (personal burnout

(60.4%), work-related burnout (50.6%), and client-related burnout (31.5%)) among HPs in Macao during non-acute phase of the pandemic. Moreover, while anxiety (60.6%) and depression (63.4%) were prevalent and, as previously reported, strongly associated with burnout. The positive associations of working in the public sector, working in the hospital setting and participating in COVID-19 duties with moderate to severe burnout were also identified. Adequate workforce management practice was found to be negatively associated the likelihood of burnout. Collectively, these findings shed light on effective interventions which support HPs' mental well-being from policy-making, organizational and induvial levels.

## Burnout is a universal concern for HPs during the COVID-19 pandemic

The findings that the burden of burnout affecting at least half of the HPs during the COVID-19 pandemic is in keeping with other recent similar studies [54]. Previously, a cross-sectional study that employed the CBI found that physicians in primary care settings in Portugal experienced higher levels of personal (65.9%), work-related (68.7%) and client-related (54.7%) burnout across different stages of the pandemic [48]. Similarly, a study in Saudi Arabia also found a higher prevalence of burnout (75%) among hospital doctors and nurses during the pandemic [55]. Related studies also reported that burnout was also common in Malaysia (personal (53.8%), work-related (39.1%) and client-related (17.4%) burnout among doctors, social workers and other types of healthcare workers [56]) and Singapore (personal burnout (49.3%) among doctors and nurses at emergency departments or urgent care centers [57]). Indeed, cross-country studies involving more than 60 countries also found that burnout among HPs ranged between 51.4% [58] and 67% [59]. This study reaffirmed that HPs were commonly under the impact of burnout in different countries during different stages of the pandemic.

## Anxiety and depression added to the concerns for HPs' mental health

A significant proportion of the respondents scored abnormal cases of anxiety (21.5%) and depression (33.3%) raising further concerns about HPs' mental health during the pandemic. This finding is consistent with a recent meta-analysis of 13 studies involving 33,062 HPs in China and Singapore, which demonstrated that approximately 1 in 5 HPs had experienced symptoms of anxiety (23.2%) or depression (22.8%) during COVID-19 [4]. This is not surprising as empirical research had already shown that during major infection outbreaks (such as the Severe Acute Respiratory Syndrome (SARS) or the Middle East Respiratory Syndrome (MERS)), it was common for HPs to experience high levels of anxiety and depressive symptoms [60, 61]. In fact, the WHO already alerted that, while the global prevalence of anxiety and depression had increased significantly by 25% in the first year of the COVID-19 pandemic [62], HPs usually experienced higher rates of anxiety and depression when compared with the general population [63, 64]. This called for closer attention to the impact of anxiety and depression on HPs' mental health for their sustainable performance in the long run of anti-COVID-19 measures.

## Factors associated with burnout among HPs

**Psychological factors.** Indeed, consistent with previous findings [65], this study found that anxiety and depression were strongly associated with moderate to severe personal burnout, work-related burnout and client-related burnout among HPs. Compared with work-related burnout and client-related burnout, abnormal cases of anxiety (OR 14.326; 95% CI 4.317–47.543) and abnormal cases of depression (OR 12.050; 95% CI 5.429–26.791) were particularly associated with personal burnout. The significant and complex burden of psychological morbidity would inevitably lead to a negative impact on individual HPs and their patient

outcomes [66, 67]. Early screening of the mental wellbeing as well as timely professional and psychological support interventions to identify and support HPs are crucial in the times of the COVID-19 pandemic [68, 69].

**Work-related factors.** It is also interesting to learn from this study that the likelihood of burnout was also associated with working in the public sector, working in hospital settings, and participating in professional duties related to anti-COVID-19 measures. Like many parts of the world, Macao lacked experiences and capacity readily deployable to deal with a pandemic like COVID-19. As such, increased workforce demands to execute public health measures inevitably necessitated reallocation and redeployment of HPs across different sectors. When the pandemic first started in early 2020 in Macao, most of the public health measures were executed upon the reallocation of health workforce from the public sector and from the hospital setting. HPs from the private sector or other non-hospital settings were later called for to supplement the manpower.

HPs with little experiences in public health were quickly assigned to COVID-19 screening and emergency team, and quarantine facilities. Uncertainties about the rapidly changing guidelines, unfamiliar high-risk settings, emerging new roles, the mounting workload and the duration of being away from their primary roles, as well as the negative impact on the team dynamic were often found concerning [70, 71]. As such, HPs were challenged not only by a novel infection but also new clinical areas or unfamiliar environment, which could easily raise psychological stress that led to increased rates of burnout [72]. Attention should be paid to the psychological state of HPs who were challenged with a constant pressure source, enduring the need to adapt or cope with a high level of uncertainties at work [73].

**Other precipitating factors.** Other precipitating factors of burnout among HPs identified in this study were in consistent with previous findings and included both demographic attributes (younger age [74], being female [75, 76], being a parent [39], being single having [77], lower education background [40] and having a history of chronic or severe diseases [41]) and professional attributes (fewer years of practice and management duties [78]). Indeed, HPs who were young, female and a parent were an important make-up in the HP workforce in Macao. HPs who were parents were prone to the fear of transmitting the infection to their children [39]. Moreover, younger HPs, especially those at a more junior level, might find it particularly challenging to adapt to new methods of working, increased service demands, prolonged periods of wearing personal protective equipment, feeling "powerless" to manage patients' conditions, and a fear of becoming infected or infecting others [78]. Measures to help alleviate burnout should be more targeted to address the needs of more vulnerable subgroups of the HP workforce.

## Implications for interventions mitigating HPs' burnout

In this study, it was found that while higher level of perceived health and COVID-19 knowledge had negative associations with moderate to severe burnout, such factors as job-related stress, high workload, an unhealthy or unsafe work environment, and insufficient organizational support were often found to be the driving forces of burnout [58]. There is a clear need for appropriate interventions to identify and manage HP's burnout as a priority [79]. At present, while evidence for specific interventions supporting HPs' mental health during public health crisis is still developing [80], various methods to prevent or reduce burnout should be carefully explored at different levels: policymaker, organization, and individual HP levels [81, 82].

**At policymaker level.** Effective interventions should create enabling practice environments where HP's burnout phenomenon can be brought to minimal. To do so, political consensus and high-level engagement in system-level actions that recognize the importance of

HP's mental health to the continuity of essential public health services and ensure HPs a safe and supportive work environment are essential to address HP's burnout [18]. Investments to change organizational behaviors and to implement workplace interventions which had significant effects on relieving workplace stress were directly associated with a reduction in the burnout scores [83–86]. As such, policymakers' awareness of burnout and commitment is important to implementing appropriate interventions for addressing burnout among the healthcare workforce. Decision-makers at the system level should be able to recognize the factors associated with burnout and be prepared to provide a capable environment as well as emotional, psychological, and financial support whenever appropriate as an appreciation message [87]. Moreover, safeguarding HPs' mental health should not be treated as an intervention strategy in a silo but be adopted with an integrated systems approach [80].

**At organization level.** One of the key organizational strategies to create a capable environment that reduces burnout was improving communication mechanisms and skills [88–90]. Uncertainty about the course of the pandemic course and unpredictability of public health measures (e.g., frequent changes of operation protocols and roles, unprecedented changes in work schedule) had been identified as contributing factor of burnout [56]. Developing clear and up-to-date guidelines and protocols for different situations, as well as practical training about protective interventions, may increase the sense of safety, assurance, and control among HPs [81, 83, 88]. In addition, HPs should be enabled to be heard, protected, prepared, and supported by their organization. Embedding access to mental health support in a safe and efficient working environment and holding workshops on coping skills for HPs involved in COVID-19 duties might also be effective in promoting collegial social support and a personal sense of control [86, 91]. Peer supervision and strong teamwork strengthening workplace well-being were also protective against burnout [92].

Other interventions such as improving workflow management, providing the opportunity to have adequate rest and exercise, arranging discussion meetings, and increasing interoperability were also worth consideration [82, 88–90]. An example of an integrative approach to mitigating burnout in HPs might involve the harnessing of the benefits of PPE accessibility and social interactions [93–95]: to install photos or at least the name tag of the staff on their PPE to promote interpersonal relations and interactions despite the difficulty of face recognition [83]. On the other hand, as the WHO has reminded [96], a lack of transparency or an imbalance between effort and reward might easily lead to feelings of injustice or incompetence, which in turn worsening HP's burnout. Improving salary scales or transparency thereof, rationalizing duty hours, creating better career opportunities, and expanding the health workforce were all expected to help reducing burnout and enhance job satisfaction of HPs in general [68].

**At individual HP level.** At the start of the COVID-19 pandemic, it was common for HPs to have an initial sense of eagerness to contribute to the healthcare effort and a sense of obligation to work through hardship [97]. Over time, the obligation to provide selfless service to the community might easily lead to neglecting their own physical, mental, social, and emotional health [87]. In addition to their professional identity, HPs also played multiple other roles, including parents, children, siblings, and friends. HPs should be mindful about protecting their own mental health and maintaining physical and emotional hygiene as an effective strategy to reduce burnout. Simple measures such as regular exercise, drinking water, and having a good rest [90], interaction with and social support by family members and loved ones are effective measures in reducing burnout [95, 98]. Promoting self-management, and learning about physical, mental, and emotional self-care would also be effective [99, 100].

## Way forward

As reminded by the World Health Organization (WHO) [2], the adverse mental effects on HPs during major public health incidents, if not dealt with appropriately, could result in detrimental consequences not only for the individual's mental health and physical wellbeing, but also for the quality of patient care they provided and the function of the health system they served [3]. Foreseeable new waves of infections caused by new variants of COVID-19 will continue to pose serious impact on the health system. HPs involved in the prevention, diagnosis and management COVID19 as well as those playing the supporting role will continue to work in overwhelmingly stressful environment and the challenge of burnout will inevitably persist. Ongoing research is warranted to gain a better understanding about the burnout phenomenon experienced by HPs and to seek effective intervention that support HPs' mental well-boing. Longitudinal quantitative studies as well as and qualitative studies that report HPs' and key stakeholders' perspectives should be employed to identify coordinated solutions at policy, organization and individual levels.

## Strengths and limitations

While most of the current literature reported about burnout among HPs at critical times of the COVID-19 pandemic, this study sought to determine the burnout during the non-acute phase of the public health challenge. The study findings reaffirm that burnout remains prevalent among HPs regardless of the phase of the pandemic. By using the standardized and validated measurement tool of the CBI and adopting a transparent protocol, this study allows cross-study comparison and provides a foundation for follow-up, long-term study. In order to further explore solutions to alleviate HPs' burnout, the current study also took reference of human resources recommendations made by the WHO and sought HPs' opinions that helped inform the prioritization of actions.

However, it is worth noting that this study is only a cross-sectional study. Due to its nature, it can only provide a snapshot of the burnout prevalence and the possible risk factors but not able to identify any causal relationship between burnout and all the variables tested. Moreover, the participants were not followed and there were no pre- and post-pandemic studies to allow comparison of the current trend against pre-pandemic baseline data. Secondly, the HADs employed in this study were mainly for screening purposes only. Although the current findings affirm previous understanding of prevalent burnout, anxiety, and depression among HPs during the COVID-19 pandemic, whether such responses relate to actual diagnosis of a mental illness warrants further investigation. Our dissemination strategy precluded a formal response rate calculation. Selection bias and response bias may have resulted in an overestimation or underestimation of psychological distress and rates of pre-existing psychiatric history. Bias towards potential volunteerism might have inevitably increased sampling error, affecting the generalizability of the study findings and restricting the inference made about the entirety of the HP population.

## Conclusions

Burnout is prevalent among the HPs during the COVID-19 pandemic and should be considered as a priority of concern due to its ongoing impact on the individual, patient care, and the function the health system. Personal and workforce management factors were found attributable to the risks of burnout, requiring attention and coordinated action from individual, organizational and policy-making levels. In particular, awareness of healthcare managers and policymakers is vital to bringing changes to mitigate HPs' burnout. Research about HPs' burnout is warranted to monitor the trend of the burnout phenomenon under the changing impact

of COVID-19, and to inform targeted interventions that bring solid improvement for the vulnerable groups of HPs.

## Acknowledgments

We thank the pharmacists for their participation in this study.

## Author Contributions

**Conceptualization:** Carolina Oi Lam Ung.

**Supervision:** Hao Hu, Anise Man Sze Wu, Carolina Oi Lam Ung.

**Validation:** Carolina Oi Lam Ung.

**Writing – original draft:** Yu Zheng, Pou Kuan Tang.

**Writing – review & editing:** Guohua Lin, Jiayu Liu, Hao Hu, Anise Man Sze Wu, Carolina Oi Lam Ung.

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
