## [Decision Letter · Decision Letter 0]

1 Dec 2022

PONE-D-22-29630Burnout among healthcare providers: Its prevalence and association with anxiety and depression during the COVID-19 pandemic in Macao, ChinaPLOS ONE

Dear Dr. Ung,

Thank you for submitting your manuscript to PLOS ONE. After careful consideration, we feel that it has merit but does not fully meet PLOS ONE’s publication criteria as it currently stands. Therefore, we invite you to submit a revised version of the manuscript that addresses the points raised during the review process.

Please submit your revised manuscript by Jan 15 2023 11:59PM. If you will need more time than this to complete your revisions, please reply to this message or contact the journal office at plosone@plos.org. Please include the following items when submitting your revised manuscript:A rebuttal letter that responds to each point raised by the academic editor and reviewer(s). You should upload this letter as a separate file labeled 'Response to Reviewers'.A marked-up copy of your manuscript that highlights changes made to the original version. You should upload this as a separate file labeled 'Revised Manuscript with Track Changes'.An unmarked version of your revised paper without tracked changes. You should upload this as a separate file labeled 'Manuscript'.

We look forward to receiving your revised manuscript.

Kind regards,

Sharon Mary Brownie

Academic Editor

PLOS ONE

Journal Requirements:

"The research was funded by the University of Macau (SRG2021-00007-ICMS)."

Reviewers' comments:

Reviewer's Responses to Questions

**Comments to the Author**

1. Is the manuscript technically sound, and do the data support the conclusions?

Reviewer #1: Yes

Reviewer #2: Yes

2. Has the statistical analysis been performed appropriately and rigorously? 

Reviewer #1: No

Reviewer #2: Yes

3. Have the authors made all data underlying the findings in their manuscript fully available?

Reviewer #1: Yes

Reviewer #2: Yes

4. Is the manuscript presented in an intelligible fashion and written in standard English?

Reviewer #1: Yes

Reviewer #2: Yes

5. Review Comments to the Author

Reviewer #1: Introduction section

• Burnout has been shown in studies to be common in HPs, and it is closely related to anxiety and depression, according to the authors (line 73-78).

• Given the high prevalence of burnout in HPs, which has been linked to anxiety and depression, the question here is, what gaps will be addressed in this study, and what makes this study unique? What new body of knowledge has been added to the global community?

•The rationale What are the gaps that need to be filled, and what makes it unique? This is not convincing and should be demonstrated clearly.

Methods sections

Study design

The study design should be rewritten succinctly —-too much information

The study design should explain why a survey is the best method of data collection for the study.

The statement regarding ethical consideration is also included in the study design section and is suggested to be separate and placed at the end of the method section.

Study setting and target population

•The target populations were all Macao-based HPs. But who exactly are the HPs?

•In which setting do HPs practice (hospital, health center...???),

•Are healthcare providers registered with the Health Bureau in Macau as of 2020 but not practicing or living in Macau included or excluded from the study??

All this suggest that the target population be clearly defined by clearly indicating who is included and who is excluded

Sampling methods and size

The author simply stated 371 as a minimum sample size, but the sampling techniques used to select the sample are unclear —- suggest that the sampling methods used to select these observational units be described.

Furthermore, a contradictory statement is written at the beginning of the methods sections: "open and online survey to be completed voluntarily by healthcare providers in Macao between October and December 2021." (line 102)

Respondents who had completed the COVID-19 duties were invited to complete the Section 4 questions. This section was created to investigate HP's perception of health workforce management during the COVID-19 pandemic and to identify areas where improvements could help to mitigate burnout (line 192-194)

• Are these study participants distinct from the 371?

• How many of these took part?

• How were these participants chosen to respond to Section 4?

The author gathers data through volunteerism (volunteer sampling) and snowballs sampling. These approaches, commonly known as convenience sampling, have the lowest credibility of any sampling method and should only be used as a last resort.

This is a bias toward potential volunteerism. The sample is unlikely to be representative of the target population, has higher selection bias, does not provide a basis for estimating sampling error, and restricts the use of inferential statistics.

Analysis

•the reason /assumption why using Spearman’s rho test are not justified

Result

The number of study participants reported in the results section is inconsistent with the calculated sample size (377) vs actual participation (498).

logistic regression analysis

• There is no evidence that the authors checked logistic regression assumptions (such as the assumption of collinear relationships among explanatory variables- correlation of |r|, VIF value, etc.) before using the fitted model to infer the relationship between the response variable and the explanatory variables.

• Furthermore, the reported OR is unclear as to whether it is log odds or adjusted (exponatiated );

• the reference category of the variables entered into the model is unknown; and

• logistic regression analysis, such as OR and CI, is not interpreted.

• the direction and the strength of correlation is not mentioned

Discussion

• Suggest to Start by restating the research question, followed by discussing the main findings and

• then interpretation/explain its implication(result (explaining the result –based on theories or researchers logical analysis .

• Compare findings in light of other, similar studies’ findings. If difference, explain the differences (methods, setting, participants)

• indicate Benefits or implications of the findings for patients, practitioners, health care organizations, researchers, or policymakers?

• State clearly what your study added to the research area and potential further research that should be conducted. (future areas of research )

• Strength and limitation

• Conclusion and Recommendation

Reviewer #2: Overall, this is an excellent article and very well written. The impact of COVID-19 on health and care workers is possibly one of the key public health issues of our time, with potential consequences that can reverberate for the next few years and decades. I would like to congratulate the authors for such a well developed and comprehensive paper that examines burnout and various determinant of burnout among HPs.

I only have two suggestions for the consideration of the authors:

1. In the Introduction, it would be good to highlight that the COVID-19 pandemic has had tremendous impacts on the physical and mental health and wellbeing of the HPs as well as on their distribution and on their working conditions. This multidimensional impact is quite extraordinary, and there is an interplay of factors which often exacerbate each other. The WHO Weekly Epidemiological Update on COVID-19 published on 30 March 2021 (https://www.who.int/publications/m/item/weekly-epidemiological-update-on-covid-19---31-march-2021) first revealed this comprehensive framework for measuring the impact of COVID-19 on HPs and this has since been tested and validated across countries of various settings (https://www.who.int/publications/i/item/9789240039001;
https://www.who.int/publications/m/item/impact-of-covid-19-on-human-resources-for-health-and-policy-response--the-case-of-belize--grenada--and-jamaica). Burnout has been highlighted as one of the impacts in the mental health dimension of the framework. So it would be ideal to include a citation for the multidimensional framework, just to sensitize the readers that burnout is just one of the multitudes of impacts of the pandemic on HPs.

2. A recent publication has looked at the mental health impact of COVID-19 on HPs. It highlights that burnout has been an issue for HPs even before the pandemic, but COVID-19 has exacerbated this further (pages 12 and 13). Section 2 presents interventions to protect the mental health of HPs at the organizational level and psychosocial interventions, while Section 3 presents policy actions and recommendations, which are very much in line with the interventions and recommendations outlined in this paper. Hence, I would suggest that you reference this report at the very least, and if needed, tweak language to be more aligned with it. The report is titled "Our duty of care: A global call to action to protect the mental health of health and care workers" and can be accessed here - https://2022.wish.org.qa/wp-content/uploads/2022/10/QFJ9259-02-Our-Duty-Of-Care-WEB.pdf

Best wishes to the authors..

6. PLOS authors have the option to publish the peer review history of their article (what does this mean?). If published, this will include your full peer review and any attached files.

Reviewer #1: No

Reviewer #2: No

---

## [Author Response · Author response to Decision Letter 0]

14 Jan 2023

Please kindly refer to our Letter of Response for all of our responses raised by the reviewers.

---

## [Decision Letter · Decision Letter 1]

6 Mar 2023

Burnout among healthcare providers: Its prevalence and association with anxiety and depression during the COVID-19 pandemic in Macao, China

PONE-D-22-29630R1

Dear Dr. Carolina Oi Lam Ung,

We’re pleased to inform you that your manuscript has been judged scientifically suitable for publication and will be formally accepted for publication once it meets all outstanding technical requirements.

Kind regards,

Sharon Mary Brownie

Academic Editor

PLOS ONE

Reviewers' comments:

Reviewer's Responses to Questions

**Comments to the Author**

1. If the authors have adequately addressed your comments raised in a previous round of review and you feel that this manuscript is now acceptable for publication, you may indicate that here to bypass the “Comments to the Author” section, enter your conflict of interest statement in the “Confidential to Editor” section, and submit your "Accept" recommendation.

Reviewer #2: All comments have been addressed

Reviewer #3: All comments have been addressed

2. Is the manuscript technically sound, and do the data support the conclusions?

Reviewer #2: Yes

Reviewer #3: Yes

3. Has the statistical analysis been performed appropriately and rigorously? 

Reviewer #2: Yes

Reviewer #3: Yes

4. Have the authors made all data underlying the findings in their manuscript fully available?

Reviewer #2: Yes

Reviewer #3: Yes

5. Is the manuscript presented in an intelligible fashion and written in standard English?

Reviewer #2: Yes

Reviewer #3: Yes

6. Review Comments to the Author

Reviewer #2: All comments have been addressed by the authors. Where needed, they have incorporated appropriate edits in the manuscript. I have no further suggestions.

Reviewer #3: relevant study , can serve as a basis for continuous improvement in work management and resource allocation

Appropriate methodology, good data analysis with visible results. well-identified strengths and limitations

Appropriate conclusions from the study.

7. PLOS authors have the option to publish the peer review history of their article (what does this mean?). If published, this will include your full peer review and any attached files.

Reviewer #2: No

Reviewer #3: No

---

## [Editor Report · Acceptance letter]

8 Mar 2023

PONE-D-22-29630R1 

Burnout among healthcare providers: Its prevalence and association with anxiety and depression during the COVID-19 pandemic in Macao, China 

Dear Dr. Ung:

I'm pleased to inform you that your manuscript has been deemed suitable for publication in PLOS ONE. Congratulations! Your manuscript is now with our production department. 

Kind regards, 

on behalf of

Professor Sharon Mary Brownie 

Academic Editor

PLOS ONE